# Data Poisoning Attack against Neural Network-Based On-Device Learning Anomaly Detector by Physical Attacks on Sensors

**DOI:** 10.3390/s24196416

**Published:** 2024-10-03

**Authors:** Takahito Ino, Kota Yoshida, Hiroki Matsutani, Takeshi Fujino

**Affiliations:** 1College of Science and Engineering, Ritsumeikan University, Kusatsu 525-8577, Japan; ri0110ii@ed.ritsumei.ac.jp (T.I.); fujino@se.ritsumei.ac.jp (T.F.); 2Faculty of Science and Technology, Keio University, Yokohama 223-8522, Japan; matutani@arc.ics.keio.ac.jp

**Keywords:** MEMS accelerometer, acoustic injection attack, Edge AI, on-device learning, data poisoning attack, concept drift

## Abstract

In this paper, we introduce a security approach for on-device learning Edge AIs designed to detect abnormal conditions in factory machines. Since Edge AIs are easily accessible by an attacker physically, there are security risks due to physical attacks. In particular, there is a concern that the attacker may tamper with the training data of the on-device learning Edge AIs to degrade the task accuracy. Few risk assessments have been reported. It is important to understand these security risks before considering countermeasures. In this paper, we demonstrate a data poisoning attack against an on-device learning Edge AI. Our attack target is an on-device learning anomaly detection system. The system adopts MEMS accelerometers to measure the vibration of factory machines and detect anomalies. The anomaly detector also adopts a concept drift detection algorithm and multiple models to accommodate multiple normal patterns. For the attack, we used a method in which measurements are tampered with by exposing the MEMS accelerometer to acoustic waves of a specific frequency. The acceleration data falsified by this method were trained on an anomaly detector, and the result was that the abnormal state could not be detected.

## 1. Introduction

The use of machine learning (ML) techniques has become widespread. Edge AIs, which directly install ML models (e.g., neural networks) on devices [1], can achieve low latency and privacy by performing inference and learning on the devices without outsourced processing. A typical implementation of Edge AIs is on-device inference, where an ML model is trained in a development (lab) environment and only performs inference on the device in the installed (field) environment. However, the task accuracy of the model can decrease when the lab and field environments differ due to characteristics involving installation locations and environmental changes over time [2], i.e., “concept drift” [3]. On-device learning Edge AI has been proposed to address this challenge [4]. It is expected to maintain inference accuracy over a long period of time by optimizing ML models in the field environment after placement.

Edge AIs are at risk of physical attacks targeting the ML models because the devices are directly installed in the operational environment [5]. In particular, on-device learning makes it more difficult to manage the training process compared to on-device inference. This makes data poisoning attacks a major threat. The data poisoning attack degrades inference accuracy by injecting malicious data into the training process. Although countermeasures against data poisoning attacks have been discussed [6,7,8], most have assumed Cloud AIs and batch (non-real-time) learning environments. Therefore, applying them to on-device learning is difficult because it assumes limited computational resources and low-latency training. Establishing an anti-poisoning method specialized for on-device learning is thus required.

There have been few risk assessments of data poisoning attacks against on-device learning. Before considering countermeasures, it is necessary to organize threat models and evaluate realistic attack threats. We previously demonstrated the risk of a data poisoning attack against an on-device learning anomaly detector [9]. In that demonstration, we adopted ONLAD (ON-device sequential Learning semi-supervised Anomaly Detector) [4] as an anomaly detection system that detects abnormal vibration observed by a MEMS accelerometer on a rotating cooling fan. During the training process, a data poisoning attack was performed by irradiating the MEMS accelerometer with acoustic waves in the audible range to tamper with the sensor’s output. We showed that while a normally trained anomaly detector could detect abnormal vibrations, a poisoned anomaly detector inferred the abnormal vibrations as “normal”.

In this paper, we demonstrate the risk of a more practical anomaly detection system based on ONLAD, which additionally includes concept drift detection [10] and multi-instance [11]. In the previous report [9], ONLAD was assumed to have only one ML model, but in this paper, we assume a multi-instance, which means it has multiple ML models (instances). Even if the equipment has multiple normal patterns, multi-instance maintains anomaly detection accuracy because instances learn normal patterns individually. In addition, by combining a lightweight concept drift detection [10] suitable for edge devices, the anomaly detection system can determine the number of instances. Attackers can carry out data poisoning attacks on anomaly detection systems to create malicious instances by tampering with sensor values. We show that the attack threatens the anomaly detection system’s ability to determine abnormal vibrations. To summarize the above, this paper reports that the new features [10,11] for a more practical anomaly detection system than our previous report create a new attack surface for the data poisoning attacks.

The main contributions of this paper are as follows.

We present a data poisoning attack scenario for an on-device learning anomaly detection system with concept drift detection and multiple detection instances.We conducted experiments based on the attack scenario outlined above to evaluate the threat. We tampered with the observed data by irradiating the accelerometer with acoustic waves and had the anomaly detection system create an instance using this as training data. We showed that this instance would determine abnormal vibrations as “normal”, making it impossible for the anomaly detection system to determine that abnormal vibrations were abnormal.

## 2. Preliminaries

### 2.1. Autoencoder-Based Anomaly Detector

The anomaly detection task distinguishes between data sampled from a distribution that is considered to be “normal” or “abnormal”. In many cases, a larger number of normal samples are available, but few or no abnormal samples are available. Therefore, a typical anomaly detector learns only the distribution of normal data and considers data that deviates from that distribution as abnormal. Methods based on the nearest neighbor [12], one-class classifier [13], and neural networks [14] have all been used for such detection.

In this paper, we focus on a neural network-based anomaly detector [14] that adopts an autoencoder [15], as shown in Figure 1. The autoencoder is roughly divided into an encoder part, which encodes the input into an intermediate vector, and a decoder part, which reconstructs the input from the intermediate vector. The intermediate vector is often smaller than the input. During the training phase of the autoencoder, the weight parameters are adjusted so that the difference between the input and output data becomes small.

An autoencoder for anomaly detection is typically trained with normal data [16,17] and can reconstruct normal data well in the inference phase. In contrast, when abnormal measurement data (abnormal data) are used as the input, the autoencoder cannot reconstruct it correctly because it has not been trained on the abnormal data. The difference between the input and output data is small for normal and large for abnormal data. We can distinguish abnormal data by setting a threshold; namely, the input is inferred to be normal if the difference is smaller than the threshold and abnormal if it is larger.

### 2.2. Concept Drift

Concept drift is a phenomenon in which the distribution of data changes over time from the training [18]. Concept drift due to changes in the surrounding environment is expected at sites where Edge AI devices are deployed [19]. In anomaly detection tasks, if normal data distribution deviates from the initial (trained) distribution due to concept drift, there is a concern that normal data may be distinguished as abnormal, thereby degrading accuracy.

### 2.3. Extreme Learning Machine

The extreme learning machine (ELM) [20] shown in Figure 2 is a feed-forward neural network consisting of an input layer, hidden layer, and output layer. It is characterized by faster weight optimization (training) than general neural networks trained using gradient descent. Let α be the joint weights between the input and hidden layers, β be the joint weights between the hidden and output layers, *b* be the bias of the hidden layer, and *G* be the activation function applied to the output of the hidden layer. Then, the output data *y* from the input data *x* is expressed as
(1)y=G(x·α+b)·β.

In ELM training, only β is updated among the three parameters α, β, and *b*, while the others are fixed at their initial values. These initial values for weights are often set similarly to those set by the initialization process of weights in neural networks. The ELM speeds up the computation by only updating the weights for β and deterministically calculating the optimal values. Let H=G(x·α+b) be the output of the hidden layer and *t* be the ground-truth label. The optimal weight $\hat{\beta }$ obtained by training is calculated as
(2)$$\hat{\beta }=H^{†}t,$$
where $H^{†}$ is the pseudo-inverse matrix of *H*.

**Figure 2 sensors-24-06416-f002:**
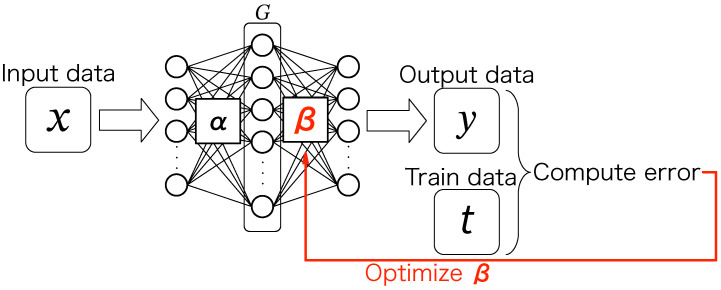
Overview of ELM.

### 2.4. Acoustic Injection Attacks on MEMS Accelerometers

MEMS accelerometers are used to measure a machine’s vibration in operation and predict machine failures [21,22]. An anomaly detector can detect abnormal vibration due to a malfunctioning component by learning the vibration of normal operation. Reports have shown that the capacitive MEMS accelerometer outputs can be falsified by irradiating acoustic waves with a specific frequency [23,24,25]. Since an embedded system blindly trusts the sensor outputs, malicious measurements can cause a system failure. Nashimoto et al. demonstrated that tilt sensor outputs with Kalman filter-based sensor fusion can be tampered with by combining multiple measurement tampering attacks, including acoustic injection.

## 3. Neural Network-Based On-Device Learning Anomaly Detector

In this paper, we adopt the on-device learning anomaly detector [11], which is a version of ONLAD [4] extended by Sunaga et al., as an evaluation target. This anomaly detector combines multi-instances for multiple concepts and a centroid-based drift detection method [10].

### 3.1. ONLAD

Typical multi-layer autoencoders and their training algorithms, such as the gradient descent method and backpropagation, are computationally intensive and require a large number of computing resources. ELM is also categorized as a batch-based algorithm. Therefore, ELMs are unsuitable for Edge AI with limited computational resources. Tsukada et al. proposed ONLAD [4], an anomaly detector that adopts online sequential ELM (OS-ELM) [26], as an autoencoder. OS-ELM is an ELM with a fast sequential learning algorithm. When the anomaly detector is trained with the *i*-th input data $x_{i}$ by OS-ELM, it computes $\beta_{i}$, which minimizes the difference between inputs and ground truth (error), as
(3)$$error=∥\begin{array}{c} \left[\begin{array}{c} H_{0}\\ \vdots \\ H_{i} \end{array}\right]\beta_{i}-\left[\begin{array}{c} t_{0}\\ \vdots \\ t_{i} \end{array}\right] \end{array}∥,$$
where $H_{i}$ is the output of the hidden layer for the *i*-th input data $x_{i}$, i.e., $H_{i}=G\left(x_{i}\cdot \alpha +b\right)$. The optimal weight $\beta_{i}$ is then calculated as
(4)$$P_{i}=P_{i-1}-P_{i-1}{H_{i}}^{T}{\left(I+H_{i}P_{i-1}{H_{i}}^{T}\right)}^{-1}H_{i}P_{i-1},$$
(5)$$\beta_{i}=\beta_{i-1}+P_{i}{H_{i}}^{T}\left(t_{i}-H_{i}\beta_{i-1}\right),$$
where the initial values $P_{0}$ and $\beta_{0}$ are defined as
(6)$$P_{0}={\left({H_{0}}^{T}H_{0}\right)}^{-1},$$
(7)$$\beta_{0}=P_{0}{H_{0}}^{T}t_{0}.$$

OS-ELM can compute the optimal weight parameter β for new training data without requiring access to past training data. Therefore, OS-ELM can perform sequential learning faster and with less memory than normal ELM and neural networks with gradient descent-based learning algorithms. This enables on-device learning for autoencoder-based anomaly detection, even on edge devices with limited computational resources. When OS-ELM is used as an autoencoder for anomaly detection, the ground truth *t* above is the input data *x*.

### 3.2. Multi-Instance

Although ELM can be implemented in a lightweight manner, its representation capacity is limited because the optimization target is only β. Therefore, there is a concern that ELMs cannot be trained well when there are multiple normal patterns (e.g., when the equipment to be monitored has multiple modes of operation). An approach using multiple small ELMs (instances) has been proposed [11,27] to learn multiple normal patterns.

With this approach, we can expect one instance to learn one normal pattern. The number of instances is the same as the number of normal patterns. All ELMs in the instances share an observed input and perform inference. The anomaly detection system expects the input to belong to a distribution of the instance that achieves the smallest error among all the instances. The instance with the smallest error is used for detection.

There are multiple approaches for determining the number of instances and training them. For example, a user can specify the number of instances and manually train each instance at the time the anomaly detector is installed. A new instance can also be trained automatically (or manually started) when concept drift is detected.

### 3.3. Concept Drift Detection

There are various approaches for concept drift detection [28,29,30,31]. However, due to batch processing and other assumptions, most of these approaches are difficult to implement in real-time on an edge device. Therefore, we adopt the concept drift detection method proposed by Yamada et al. [10], which is designed for edge devices. It detects the concept drift by monitoring the centroid of the input data.

Figure 3 shows an overview of the concept drift detection, which is performed as follows. First, the centroid of training data $C_{train}$ is sequentially acquired during training (Figure 3a) by
(8)$$C_{train}=\frac{C_{train}\times N_{train}+x_{train}}{N_{train}+1},$$
where $N_{train}$ is the number of samples trained (number of centroid updates), and $x_{train}$ is the latest training data.

After the training, the centroid of the test data, $C_{test}$, is calculated sequentially (Figure 3b), as
(9)$$C_{test}=\frac{C_{test}\times N_{test}+x_{test}}{N_{test}+1},$$
where $x_{test}$ is the latest input data during inference, and $N_{test}$ is the number of updates to the centroid. The $N_{test}$ is incremented for each update. Thus, the trained and test centroids can be calculated with a small memory.

The drift rate is defined as the distance θ between the training centroid $C_{train}$ and the test centroid $C_{test}$. It is normalized so that the distance between the trained centroid and the last piece of training data is 1.0. When concept drift occurs for the test data, the distance between the trained centroid and the test centroid gradually increases (Figure 3c). Concept drift is detected when θ exceeds a predefined threshold $\theta_{drift}$. Users can set the threshold $\theta_{drift}$ to an appropriate value depending on the task. Once the concept drift is detected, the anomaly detector can follow environmental changes by re-training instances or launching new instances (Figure 3d). The drift rate is calculated for each instance when the anomaly detection system is operated with multiple instances. When a new input is given, the drift rate to be updated is that of the instance that shows the smallest error with the input data.

An example of the expected drift rate is shown in Figure 4. Here, the drift rate is within the range [0,∞] but is set to −1 as a constant when calculating the initial value of the centroid. The drift rate is 1.0 just after training, and thereafter, when no drift occurs, it is around 1.0. On the other hand, when drift occurs, the drift rate increases significantly and exceeds the threshold value $\theta_{drift}$, which is detected as concept drift.

### 3.4. Behavior of Anomaly Detector

In this paper, we assume an anomaly detector is designed to operate in two phases: training and inference. The training phase is a period for training instances in the anomaly detector. A user manually launches the training phase after the device deployment and operates equipment to be monitored in various modes. During the training phase, the anomaly detector performs training, and a new instance is automatically set when the concept drift (new normal behavior) is detected. The equipment is expected to operate normally during this phase. The inference phase is a period for detecting abnormal behavior and is launched after the training phase. Trained instances are utilized to detect anomalies.

An example of the behavior of the anomaly detector is shown in Figure 5, where we assume there are two types of normal behavior of the equipment: normal state 1 and normal state 2. First, a user launches the training phase, and normal state 1 is observed and a first instance is trained. After that, normal state 2 is observed. Since normal state 2 behaves differently from normal state 1, a concept drift is detected, and a new instance is trained. Note that all the user has to do is start the training phase of the anomaly detector and operate the equipment with two normal modes for a certain period; it is not necessary to monitor or lead the training process of the anomaly detector. The user switches to the inference phase, and samples with normal state 1 are observed. The observed samples are fed into the autoencoder for each instance, and the instance with a lower error is selected and compared to the threshold value $\theta_{error}$, which is set to detect an anomaly. Now, the observed samples are from normal state 1, so the error from the instance trained with normal state 1 is chosen, and the prediction result is “normal”. Similarly, next, the samples from normal state 2 are observed, and the anomaly detector predicts “normal”. Finally, the samples from the abnormal state are observed. Since neither of the instances has learned the abnormal state, they output the higher error. The smallest error is higher than the ones in the normal state, and of course, it is larger than the threshold $\theta_{error}$, and the anomaly detector predicts them as “abnormal”. The threshold $\theta_{error}$ is defined at the first instance that is trained. Typically, the threshold is assumed to be set outside the normal data distribution with a margin. The error is calculated as error by the root mean square error (RMSE) between the input data *x* and the output of the autoencoder.

## 4. Threat Model

### 4.1. Attack Scenario

In this paper, we define anomaly detector, victim and attacker as follows.

Anomaly detector is installed on a factory machine. The anomaly detector measures the vibrations of the machine with a MEMS accelerometer and detects anomalies. Unlike our previous report [9], the anomaly detector in this paper newly adopts a concept drift detection and multi-instance. The addition of these two functions enables the anomaly detector to accommodate multiple normal patterns.Victim (user of the anomaly detector) aims to detect abnormal behavior of equipment to be monitored by using an on-device learning anomaly detector. The anomaly detector is installed in the target equipment, and the prediction results, “normal” or “abnormal”, are checked. An accelerometer in the detector observes the vibration of the equipment during its operation. The victim does not check the data acquired by the accelerometer and the anomaly detector’s training status.Attacker aims to hide an abnormal behavior of the target equipment from the anomaly detector through a data poisoning attack. To achieve this, the attacker uses an acoustic injection attack to tamper with the accelerometer’s observation. The attacker carefully imitates the abnormal vibrations and trains the anomaly detector on it. The victim is unable to notice this because the acoustic injection attack is not an invasive attack and the acoustic wave cannot be heard by humans when it is in the ultrasonic range.

As explained in Section 3.4, adopting two new functions (concept drift detection and multi-instance) makes the anomaly detector more practical than the one in our previous report. On the other hand, we report that these functions unintentionally cause the risk of security threats. There is a threat of attacks that can make the anomaly detector unable to detect specific abnormal vibrations. The attacker tampers with the measurement values of the anomaly detector during the training phase and trains instances with malicious data. At this time, the victim cannot check the training status of the anomaly detector and is, therefore, unable to notice the existence of instances that have been trained with malicious data.

### 4.2. Attack Procedure

First, we describe an anomaly detector’s behavior without attacks, as shown in Figure 6. The victim installs the anomaly detector on the equipment to be monitored and sets it to the training phase. The victim runs the equipment in a normal state and lets the anomaly detector train an instance (normal instance) on the normal state. To simplify the scenario, we assume only one normal state in this paper. After completing the training phase, the victim moves the anomaly detector to the inference phase. Since the error of the normal instance on the normal state is small, the equipment’s normal operation can be predicted as “normal”. The error from the normal instance on the abnormal state becomes large. When the error exceeds the threshold, the anomaly detector predicts “abnormal”.

Next, Figure 7 shows our attack procedure. An attacker performs an acoustic injection attack and mimics an abnormal vibration during the training phase. The equipment operates in the normal state throughout the training phase, but the anomaly detector’s observation is tampered with during the attack. The anomaly detector detects concept drift through the attack, and a new instance (poisoned instance) is trained on the attacked (poisoned) state. Here, the anomaly detector trains two instances internally: normal and poisoned. In the inference phase, the anomaly detector predicts the normal state as “normal” because the error from the normal instance is smaller than the threshold. Similarly, the anomaly detector predicts the abnormal state as “normal” because the error from the poisoned instance is smaller than the threshold (note that the poisoned instance was trained under the attack, which carefully mimicked the abnormal vibrations). This indicates that the attacker has succeeded in hiding the abnormal state from the victim. In this way, the attacker exploits the drift detection function and creates the malicious (poisoned) instance. This is a new threat caused by adopting these features.

## 5. Experimental Setup

The experimental equipment we used is listed in Table 1. Figure 8 shows a photograph of the experimental setup, and Figure 9 shows its block diagram. Figure 8b shows a close-up view of the target equipment (cooling fan), with the accelerometer installed on top. For the cooling fan, two identical products were glued vertically so that vibrations could be transmitted to each other. This is because individual differences often exist even among the same products in the MEMS accelerometer. We constructed an experimental setup to observe both vibrations with one sensor. We attached a weight to one of the wings on the top fan. Since this caused abnormal vibration for the top fan, we considered the bottom fan normal and the top fan abnormal. Here, the state in which only the normal fan is rotating is defined as a “normal state”. The state in which only the abnormal fan is rotating is defined as an “abnormal state”. The aim of the anomaly detector is to correctly detect this vibration as abnormal.

We adopted a MEMS three-axis accelerometer (ADXL345) as the victim accelerometer. It observes vibrations generated by normal and abnormal cooling fans. The accelerometer is connected to a controller (Raspberry Pi Pico in Section 6 and Raspberry Pi4 Model B in Section 7) via an Inter-Integrated Circuit (I2C) bus. The controller collected measurement values and stored them in the PC. The sampling rate of the measurement is 100 Hz. In this experiment, we use the *x*-axis acceleration among the three axes of the accelerometer for anomaly detection. We used different controllers in the two experiments (in Section 6 and Section 7), but their settings were the same when collecting measurement values. The observed acceleration is converted into a frequency spectrum by fast Fourier transform (FFT) in an 800-point sampling window and input to the anomaly detector. FFT and anomaly detection processing were performed on the PC.

The attacker injects acoustic waves from the upper speaker into the accelerometer. A function generator is connected to the speaker to generate a sine wave at a specific output voltage and frequency. In Section 6, we additionally use an audio amplifier to amplify the signal to the ultrasonic speaker.

Table 2 lists the hyperparameters of the ELM model. These parameters were determined by searching multiple patterns, enabling the correct separation of normal and abnormal conditions and obtaining a real-time performance.

## 6. Acoustic Injection Attack against MEMS Accelerometer

We evaluated the effect of acoustic/ultrasonic wave injection on the measured values of the MEMS accelerometer ADXL345. This preliminary investigation shows that an attacker can mimic abnormal vibrations (explored further in Section 7).

We recorded 1000 samples under the following attack conditions.

Experiment to evaluate the effect of the frequency of acoustic waves: A function generator generates signals while sweeping the frequency. The signal is input to a speaker, and the generated acoustic waves are irradiated into an accelerometer. First, a full-range speaker is used to evaluate the effect of injecting audible acoustic waves (2700–3200 Hz). Next, we evaluate the effect of injecting ultrasonic waves (25,300–25,800 Hz) using an ultrasonic speaker. The output voltage of the function generator was set to the maximum allowable voltage of each speaker (P800K: 10 V, CUSA-T601-150-2400-TH: 30 V).Experiment to evaluate the effect of sound pressure: The function generator generates signals while sweeping the output voltage. The signal is input to the speaker, and the generated acoustic wave is injected into the accelerometer. The frequency of the injected acoustic waves is set to 3000 Hz with the full-range speaker. The signal voltage input to the speaker is increased from 2 V to 10 V.

Figure 10 shows the frequency spectrum of the data acquired from the accelerometer when both the cooling fans are stopped, where the horizontal axis indicates frequency and the vertical axis indicates amplitude. When we checked 100 samples acquired under the same conditions, the amplitude was below 0.015 at points other than 0 Hz. This suggests that the accelerometer is affected by acoustic waves when the amplitude exceeds 0.015 under the attack.

For the first experiment, Figure 11 and Figure 12 show heat maps summarizing the second peak in the frequency spectrum under irradiating audible acoustic waves (2700–3200 Hz) and under irradiating ultrasonic waves (25,300–25,800 Hz), respectively. The horizontal axis is the frequency of the injected acoustic wave, the vertical axis is the frequency of the second peak occurring by the attack, and the color map represents the amplitude of the peak. The second peak occurs at a frequency corresponding to the frequency of the irradiating acoustic waves.

For the second experiment, Figure 13 shows the observed frequency spectrum when the input voltage of the speaker (sound pressure) is increased. Here, the horizontal axis indicates frequency and the vertical axis indicates amplitude. The second highest peak occurred at around 11 Hz by the attack (the first was at 0 Hz). As the signal voltage input to the speaker increases, the peak amplitude increases. The peak’s frequency of occurrence hardly changes, even when the sound pressure changes.

These results demonstrate that the attacker can choose the frequency and amplitude of the peak by adjusting the frequency and amplitude of the irradiating acoustic waves.

## 7. Data Poisoning Attack against Anomaly Detector

In this section, we describe the result of a data poisoning attack on a multi-instance anomaly detector. Our attack is evaluated using scenarios described in Figure 6 and Figure 7. This section is separated into three parts. First, we show the acceleration data collected in normal and abnormal conditions. We introduce an attacker’s setup to mimic abnormal acceleration by the acoustic injection attack and show the collected acceleration data. Next, we show the behavior of the multi-instance anomaly detector under the no-attack scenario (Figure 6). Finally, we show the behavior under the attack scenario (Figure 7).

### 7.1. Frequency Spectrums of Accelerometers in Each Fan State

Figure 14 shows a sample of collected acceleration data as a frequency spectrum. First, we acquired acceleration data in the normal (Figure 14a) and abnormal (Figure 14b) states described in Section 5. The difference between the frequency spectrum of these states is an abnormal acceleration peak at 2.88 Hz due to the vibration caused by the weight attached to the abnormal fan’s blade.

As described in Section 4 (especially in Figure 7), an attacker’s objective is to make the anomaly detector have a new instance trained with acceleration similar to the abnormal state. This instance is expected to output low error in the abnormal state. The attacker irradiates acoustic waves to the cooling fan in the normal state (Figure 14a) during the training phase and mimics the abnormal state (Figure 14b).

The results in Section 6 show that the attacker can reproduce the abnormal acceleration peak by adjusting the frequency and sound pressure of the irradiating acoustic waves. We set the output frequency to 2806.6 Hz and the voltage to 9.91 V on the function generator and acquired acceleration data under the attack (the results in Section 6 show that the accelerometer is more sensitive to the acoustic waves in the audible range than the ultrasonic range; therefore, we set the function generator’s output frequency in the audible range, but of course, this attack can also be performed using the ultrasonic range to hide from the victim). Figure 14c shows the frequency spectrum of the attacked (poisoned) acceleration data, and it is similar to the acceleration in the abnormal state.

### 7.2. Behavior of Anomaly Detector in No-Attack Scenario

We show the anomaly detector’s behavior in Figure 15, which was evaluated under the no-attack scenario shown in Figure 6. The figure shows the anomaly detector’s behavior and states of experimental environments. The two graphs on the top show the anomaly detector’s error value and drift rate. The threshold $\theta_{error}$ is defined at the first instance that is trained. Typically, the threshold is assumed to be set outside the normal data distribution with a margin; we manually decided on a provisional threshold ($\theta_{error}=4.25\times 10^{-10}$) for the sake of convenience in our verification process. In actual operation, the threshold should be decided automatically based on rules selected by the developer. The “fan state” shows the state of the observation target equipment; the normal state is only the normal fan rotation, and the abnormal state is only the abnormal fan rotation. The “speaker state” shows the state of the attacker’s speaker; it does not activate in this experiment because it is evaluated under the no-attack scenario. The “primary instance” shows an instance that outputs the smallest error among all the instances. The primary instance’s error is compared with the threshold $\theta_{error}$ to distinguish whether the equipment’s state is normal or abnormal, and the detection result is shown in the “inference result”.

The anomaly detector’s behavior is different in the training and inference phase. In the training phase, normal data are expected to be observed, and the anomaly detector trains a normal instance, where the normal instance means the instance is trained with observed acceleration data in the normal state. Earlier in the inference phase, normal data are observed. The prediction error of the normal instance is lower than the threshold $\theta_{error}$, and the anomaly detector outputs the decision as “normal”. After that, abnormal data are observed. The prediction error of the normal instance is higher than the threshold $\theta_{error}$, and the anomaly detector outputs the decision as “abnormal”. This behavior shows the anomaly detector correctly detects the target equipment’s “abnormal” state.

### 7.3. Behavior of Anomaly Detector in Data Poisoning Attack Scenario

We show the anomaly detector’s behavior in Figure 16, which was evaluated under the poisoning attack scenario shown in Figure 7. The figure shows the anomaly detector’s behavior in the same way as Figure 6. The error threshold is set as $\theta_{error}=4.25\times 10^{-10}$—the same as the previous experiment. The manufacturer sets a threshold value for drift detection $\theta_{drift}$ by considering the target task; we set $\theta_{drift}=2.0$.

In the training phase, normal data are expected to be observed, and the anomaly detector trains a normal instance. After that, the attacker activates the attack speaker, and the anomaly detector observes acceleration from the poisoned state. The drift rate gradually increases after the observed acceleration data changes to the poisoned state. A concept drift is detected at the drift rate higher than the threshold $\theta_{drift}$. After the concept drift is detected, the anomaly detector makes a new instance (poison instance), trained with the acceleration data observed in the poisoned state. The attacker turns off the speaker after the poison instance is trained. The anomaly detector has normal and poison instances after the training phase. Earlier in the inference phase, normal data are observed. The prediction error of the normal instance is lower than the poison instance, which is also lower than the threshold $\theta_{error}$, and the anomaly detector outputs the decision as “normal”. After that, abnormal data are observed. The prediction error of the poison instance is lower than the normal instance, and it is also lower than the threshold $\theta_{error}$. The anomaly detector outputs the decision as “normal”. This behavior shows the anomaly detector does not detect the target equipment’s “abnormal” state.

## 8. Conclusions

In this paper, we conducted a security risk assessment of on-device learning Edge AIs. We selected the ONLAD-based anomaly detector for the evaluation target. The ELM, a lightweight training algorithm for NNs adopted in the ONLAD, enables the training of autoencoders on resource-limited edge devices. In addition, the ONLAD adopted the concept drift detection and multi-instance prediction for targeting more practical anomaly detectors. We conducted a data poisoning attack against the anomaly detector using an acoustic wave injection attack on a MEMS accelerometer.

We performed an experiment to tamper with observed acceleration by injecting audible acoustic and ultrasonic waves into a MEMS accelerometer. In this experiment, we showed that an attacker can generate a peak in a specific frequency and amplitude on the frequency spectrum. Based on these results, we reproduced data observed from the abnormal state by injecting acoustic waves into the normal state. An anomaly detector that is trained with ideal normal data could detect normal and abnormal states correctly. On the other hand, an anomaly detector that is trained with attack data detects the abnormal state as “normal”. In other words, this attack prevents the anomaly detector from detecting the specific anomaly intended by the attacker. This result demonstrated the threat of data poisoning attacks against on-device learning Edge AIs. Our experiments considered the case with a single normal state, but of course, the anomaly detector will work even when there are multiple normal states. In addition, the attack discussed in this paper is effective regardless of the number of normal states. The attacker only needs a malicious instance in addition to normal instances.

We demonstrated a data poisoning attack with one of the typical cases of abnormal data. In the future, we plan to demonstrate this attack on other abnormal detection setups [32,33,34] to show this threat applies in various cases.

The threat of data poisoning attacks by tampering with input data is not limited to accelerometers; indeed, it is expected to apply to all systems that use sensors. Similar attacks threaten systems that blindly trust sensor data, like the anomaly detection system we targeted. Considering the anomaly detector for factory machines, it is a concern that the early detection of anomalies in machines may be hindered, which may degrade maintainability and productivity. This attack threatens only on-device learning Edge AIs because it assumes the attacker can create poison instances by exploiting on-device learning functions; it can be targeted in NN models other than ELM. Since users and developers cannot always control the training data in on-device learning Edge AIs, developing effective countermeasures against such attacks is required.

## Figures and Tables

**Figure 1 sensors-24-06416-f001:**
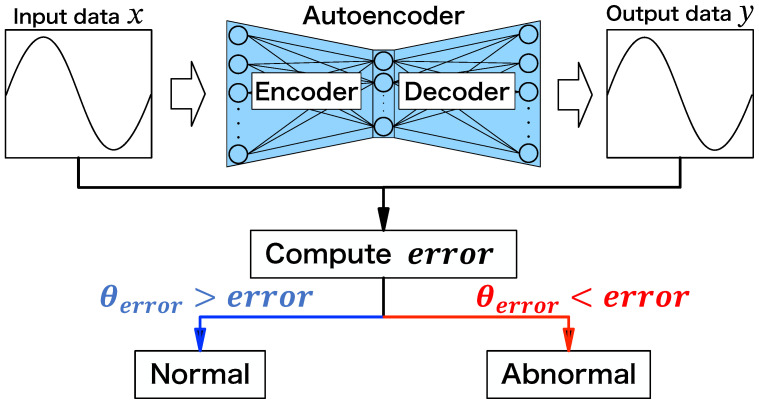
Autoencoder-based anomaly detector.

**Figure 3 sensors-24-06416-f003:**
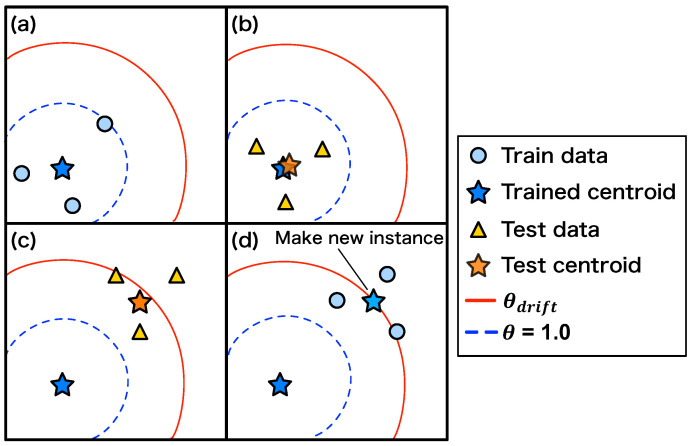
Overview of concept drift detection algorithm. (**a**) Trained centroids are sequentially calculated during training. (**b**) Test centroids are sequentially calculated during inference. (**c**) When concept drift occurs, the test centroid moves away from the train centroid. (**d**) When the test centroid exceeds the threshold, a concept drift is detected and a new instance is created. The new instance computes its own train centroid from the latest data (training data).

**Figure 4 sensors-24-06416-f004:**
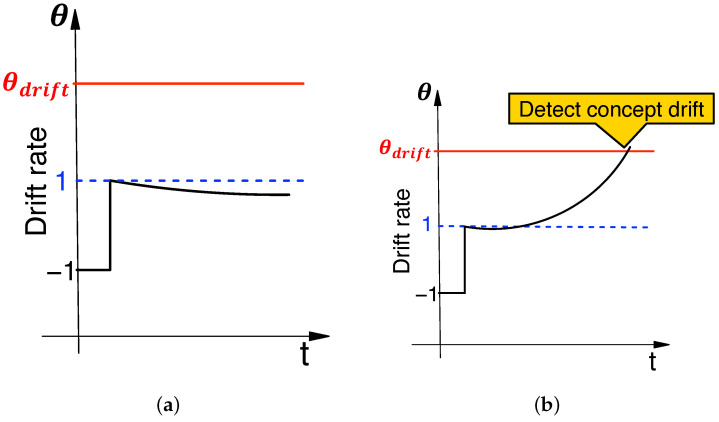
Expected drift rate behavior. (**a**) Concept drift does not occur; (**b**) Concept drift occurs.

**Figure 5 sensors-24-06416-f005:**
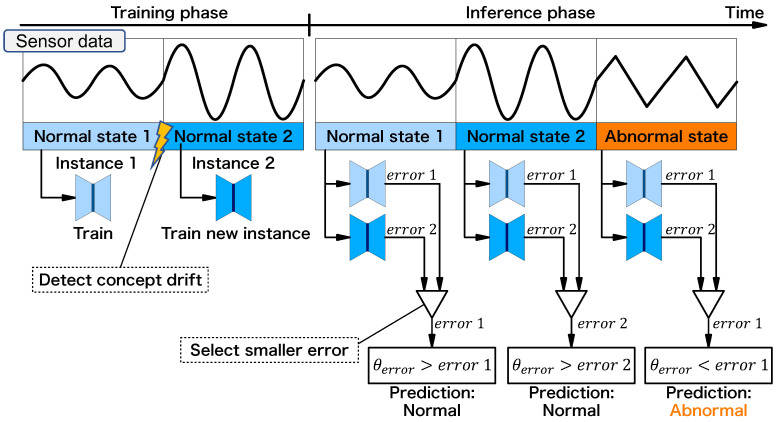
Behavior of multi-instance on-device learning anomaly detector.

**Figure 6 sensors-24-06416-f006:**
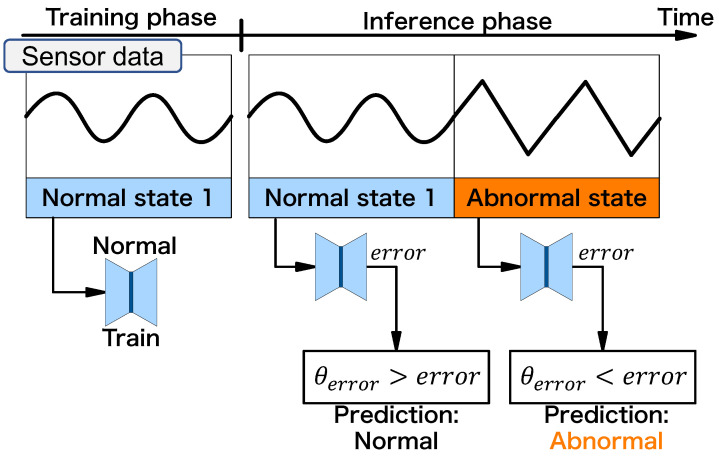
Behavior of anomaly detector without attack.

**Figure 7 sensors-24-06416-f007:**
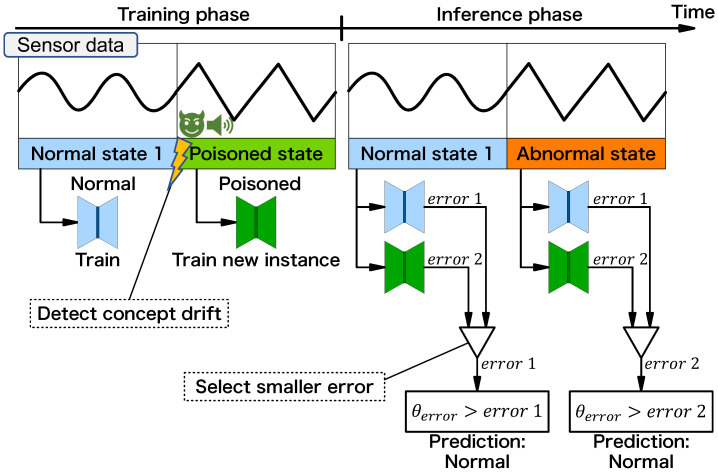
Behavior of anomaly detector with data poisoning attack.

**Figure 8 sensors-24-06416-f008:**
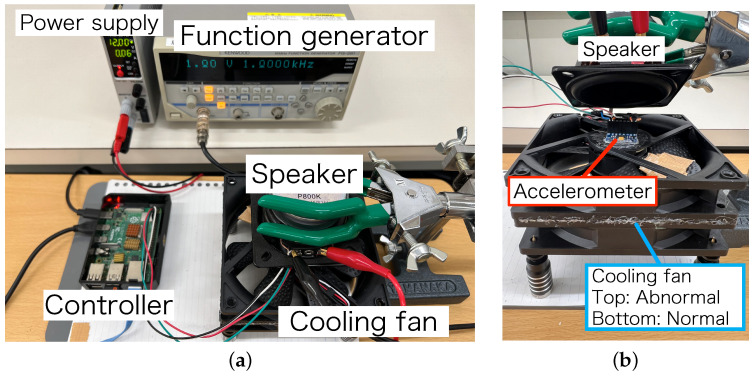
Experimental setup. (**a**) Overall setup; (**b**) Cooling fan and speaker.

**Figure 9 sensors-24-06416-f009:**
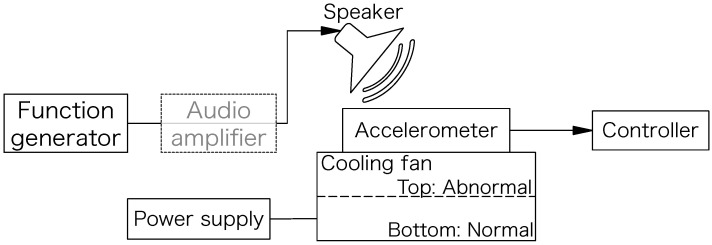
Block diagram of experimental setup.

**Figure 10 sensors-24-06416-f010:**
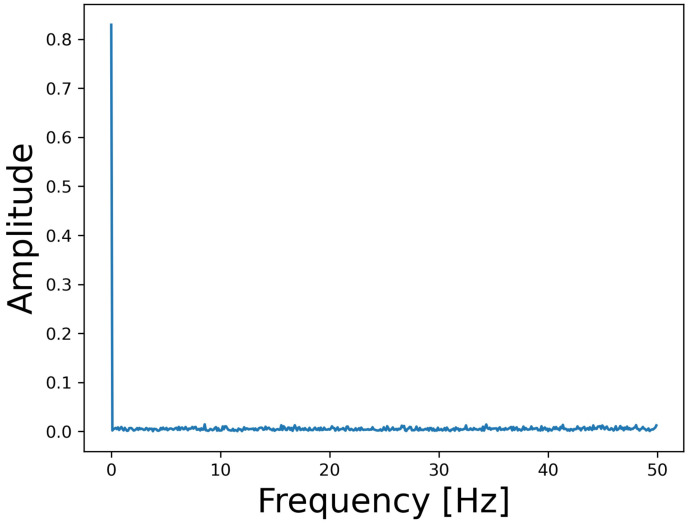
Observed frequency spectrum while both cooling fans are stopped.

**Figure 11 sensors-24-06416-f011:**
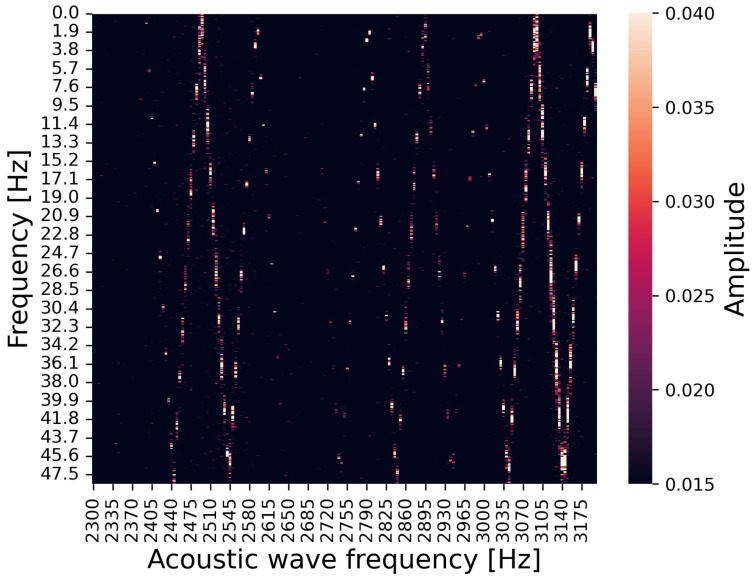
Relationship between irradiated acoustic wave frequency (in audible range), observed peak frequency, and amplitude.

**Figure 12 sensors-24-06416-f012:**
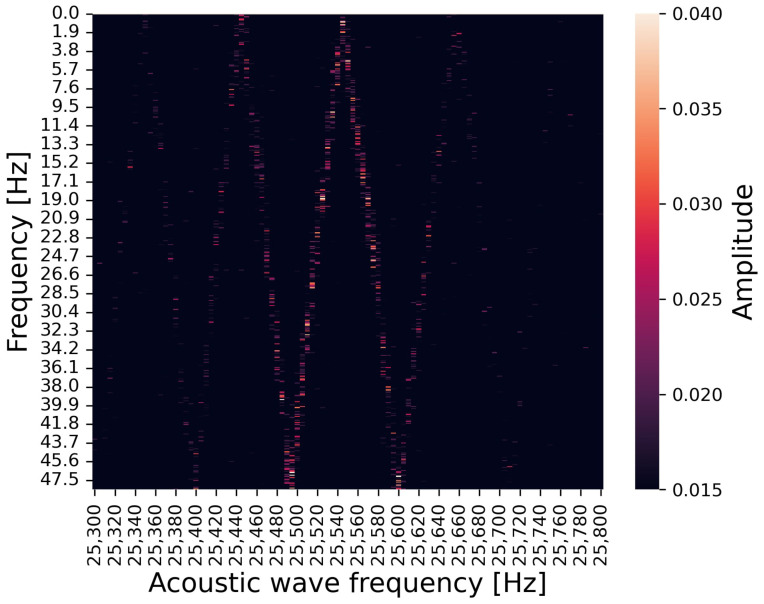
Relationship between irradiated acoustic wave frequency (in ultrasonic range), observed peak frequency, and amplitude.

**Figure 13 sensors-24-06416-f013:**
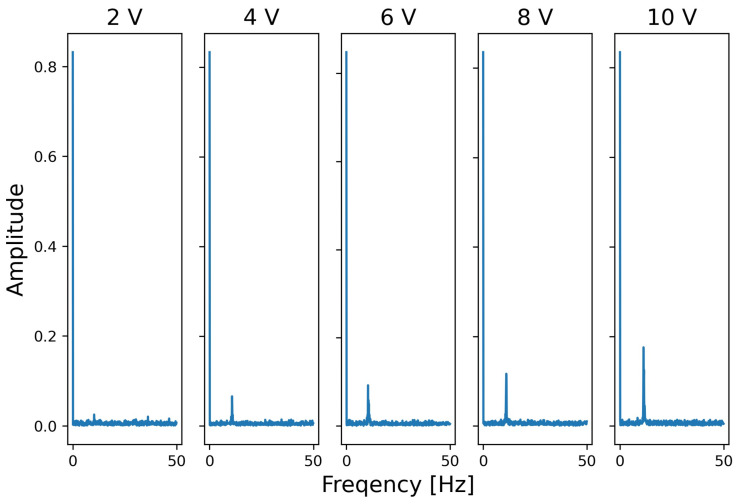
Effects of sound pressure for observed peak amplitude (frequency of acoustic waves: 3000 Hz).

**Figure 14 sensors-24-06416-f014:**
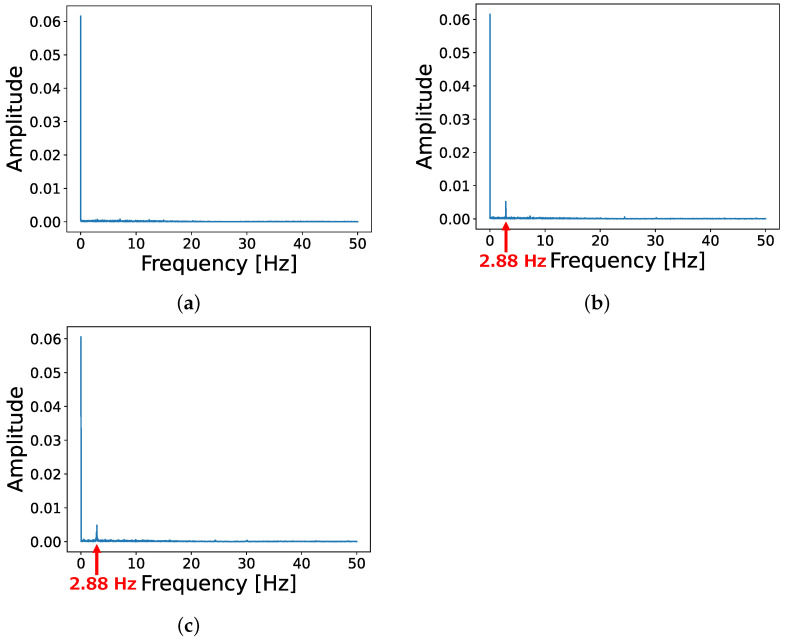
Samples of observed data. (**a**) Normal state; (**b**) Abnormal state; (**c**) Poisoned state.

**Figure 15 sensors-24-06416-f015:**
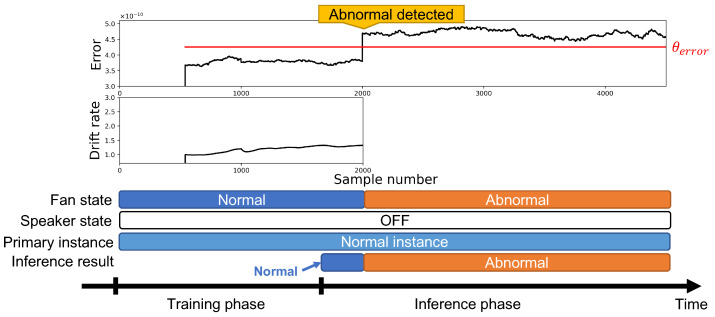
Error and drift rate without data poisoning attack.

**Figure 16 sensors-24-06416-f016:**
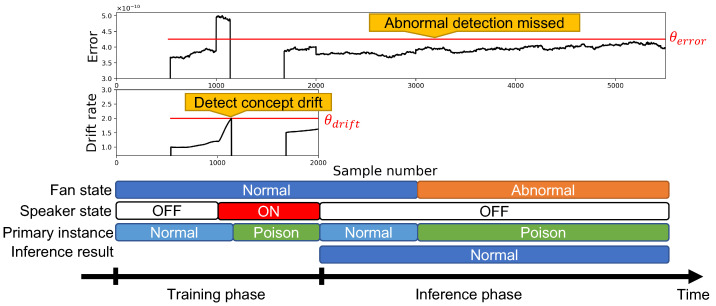
Error and drift rate with data poisoning attack.

**Table 1 sensors-24-06416-t001:** Experimental equipment.

Equipment	Model Number	Manufacturer
Function generator (in Section 6)	MFG-2260MRA	TEXIO TECHNOLOGY Corp. (Yokohama, Japan)
Function generator (in Section 7)	FG-281	JVCKENWOOD Corp. (Yokohama, Japan)
Audio amplifier	PMA-600NE	D&M Holdings Inc. (Kawasaki, Japan)
Full-range speaker	P800K	Foster Electric Co. (Akishima, Japan)
Ultrasonic transmitter	CUSA-T601-150-2400-TH	CUI Devices (Lake Oswego, OR, US)
Accelerometer	ADXL345	Analog Devices, Inc. (Wilmington, MA, US)
Controller (in Section 6)	Raspberry Pi Pico	Raspberry Pi Foundation (Cambridge, England)
Controller (in Section 7)	Raspberry Pi 4 Model B	Raspberry Pi Foundation (Cambridge, England)
Cooling fan	CFZ-120F	Ainex Co. (Higashifushimi, Japan)
Power supply	P4K-80L	Matsusada Precision Inc. (Kusatsu, Japan)

**Table 2 sensors-24-06416-t002:** ELM model configuration.

Parameter	Value
No. of input/output layer nodes	800
No. of hidden layer nodes	10
Activation	Sigmoid

## Data Availability

Raw data were generated at Ritsumeikan University. Derived data supporting the findings of this study are available from the corresponding author K.Y. upon request.

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
