# Peer review of "Data Poisoning Attack against Neural Network-Based On-Device Learning Anomaly Detector by Physical Attacks on Sensors"

_sensors, 2024, doi:10.3390/s24196416_

Round 1

Reviewer 1 Report

Comments and Suggestions for Authors

Strengths

1.  The paper implements a neural network-based anomaly detector for data poisoning attacks on devices for real-world application scenarios and conducts physical experiments for verification.

Weakness

1.  The paper is not innovative enough and is a simple application of poisoning attacks in real scenarios.

2.  The paper introduces the relevant contents of the attack scenarios at great length, and the description of the proposed method is not exhaustive enough to analyze the principle of the attack or more complex application scenarios in depth, so it is suggested to increase the analysis of the attack principle.

3.  The experiment is a single attack scenario and does not consider other anomalies to show the universality of the method, and it is suggested to conduct more experiments.

Comments on the Quality of English Language

Minor editing of English language required.

Author Response

Thank you very much for providing important comments. We are very grateful to you for spending time on our paper. We list the responses to your comments below.  

Comments 1: The paper is not innovative enough and is a simple application of poisoning attacks in real scenarios.

Response 1: To the best of our knowledge, there are few studies on security on on-device learning edge AI, especially considering physical attacks against sensors. We believe this paper's important contribution is to demonstrate that the new features (concept drift detection and multi-instance) for a more practical anomaly detection system cause a new attack surface for data poisoning attacks. This paper will encourage the study of security on on-device learning edge AIs. We inserted the summarized sentence described on L.59-61 in our manuscript to clarify our contributions.  

Comments 2: The paper introduces the relevant contents of the attack scenarios at great length, and the description of the proposed method is not exhaustive enough to analyze the principle of the attack or more complex application scenarios in depth, so it is suggested to increase the analysis of the attack principle.

Response 2: We apologize for the mixed description of the attack scenario and procedures, which made it difficult to understand the principles of the attack. We have restructured Sec. 4 to separate the scenario and procedures and clarify the principles of the attack.  

Comments 3: The experiment is a single attack scenario and does not consider other anomalies to show the universality of the method, and it is suggested to conduct more experiments.

Response 3: We demonstrated a security threat on one of the typical setups of anomaly detection tasks. In the future, we plan to demonstrate this attack on other anomaly detection setups to show this threat applies in various cases. We added a description of this limitation in conclusion (L.437-439).

Reviewer 2 Report

Comments and Suggestions for Authors

Edge AI systems face the threat of physical assaults aimed at the machine learning models, as these devices are integrated directly into their operational settings. Particularly, on-device training complicates the management of the learning process when compared to on-device inference. This difficulty elevates the risk of data poisoning attacks, a significant danger that undermines inference performance by inserting harmful data during training. It is widely understood that cloud-based AIs and batch (non-instantaneous) learning environments perform differently. Consequently, implementing them in on-device learning poses challenges due to the reliance on limited computational capabilities and a requirement for low-latency training. Therefore, there is a pressing need to develop a specialized anti-poisoning strategy tailored for on-device learning.

In their previous paper, the authors illustrated the vulnerability of data poisoning attacks targeting an on-device learning anomaly detection system. In this example, the authors utilized ONLAD (ON-device sequential Learning semi-supervised Anomaly Detector) as the mechanism for identifying unusual vibrations detected by a MEMS accelerometer situated on a rotating cooling fan. During the training phase, a data poisoning assault was executed by bombarding the MEMS accelerometer with audible acoustic waves to interfere with the sensor’s readings. The authors demonstrated that while a conventionally trained anomaly detector could identify unusual vibrations, a compromised anomaly detector misclassified these irregularities as 'normal.'

In this study, the authors reveal the dangers associated with a more practical anomaly detection framework built upon ONLAD, which now also incorporates concept drift detection and multi-instance learning. In the prior paper, ONLAD was presumed to consist of only a single ML model; however, in this document, the authors propose a multi-instance framework, meaning it operates with several ML models (instances). Despite the existence of multiple normal patterns, the multi-instance approach preserves the accuracy of anomaly detection since each instance learns normal patterns independently. Furthermore, by integrating a lightweight concept drift detection method suitable for edge devices, the anomaly detection system can regulate the number of instances. Attackers may execute data poisoning strategies on anomaly detection frameworks to fabricate malicious instances via manipulation of sensor readings. The authors demonstrate that this attack jeopardizes the system’s capacity to recognize abnormal vibrations.

Consequently, the principal contributions of this paper are as follows:

1. The authors introduce a data poisoning attack scenario for an on-device learning anomaly detection framework featuring concept drift detection and multiple detection instances.

2. The authors executed experiments based on the aforementioned attack scenario to assess the potential threat. They interfered with the collected data by bombarding the accelerometer with acoustic waves and allowed the anomaly detection system to generate an instance using this manipulated training data. The results indicated that this instance would classify abnormal vibrations as 'normal,' rendering it impossible for the anomaly detection system to identify these irregularities accurately.

The article may be published in  present form.

Author Response

We are very grateful to you for spending time on our paper. We are pleased that our paper has been reviewed to be sufficient for publication.

Reviewer 3 Report

Comments and Suggestions for Authors

Some sentences are quite complex. Breaking them down into simpler, shorter sentences will improve readability. For example, instead of saying, "Currently, as few risk assessments have been reported, it is necessary to clarify the security risks as a preliminary step in considering countermeasures," it could be simplified to, "Few risk assessments have been reported. It is important to understand these security risks before considering countermeasures."

  • The conclusion should concisely restate the main results of the experiment and their implications. It might help to specify how significant the impact of the data poisoning attack was on the anomaly detector's performance. This would strengthen the message about the seriousness of the threat.

While the conclusion touches on the broader applicability of the findings to all systems that use sensors, it could benefit from a more explicit statement about the importance of securing these systems. Highlighting the potential real-world impacts of such security breaches would emphasize the urgency for developing countermeasures.

Author Response

Thank you very much for providing important comments. We are very grateful to you for spending time on our paper. We list the responses to your comments below.

Comments 1: Some sentences are quite complex. Breaking them down into simpler, shorter sentences will improve readability. For example, instead of saying, "Currently, as few risk assessments have been reported, it is necessary to clarify the security risks as a preliminary step in considering countermeasures," it could be simplified to, "Few risk assessments have been reported. It is important to understand these security risks before considering countermeasures."

Response 1: Thank you for suggesting the clarification of the sentence. I reviewed and revised not only the sentence you mentioned but also the entire paper.  

Comments 2: The conclusion should concisely restate the main results of the experiment and their implications. It might help to specify how significant the impact of the data poisoning attack was on the anomaly detector's performance. This would strengthen the message about the seriousness of the threat.

Response 2: The significant impact on the anomaly detector's performance is that it can no longer detect the specific anomalies intended by the attacker. We added a concise description of our results in the conclusion (L.431-435).

Comments 3: While the conclusion touches on the broader applicability of the findings to all systems that use sensors, it could benefit from a more explicit statement about the importance of securing these systems. Highlighting the potential real-world impacts of such security breaches would emphasize the urgency for developing countermeasures.

Response 3: We added a description of the concerns this threat raises in real-world applications (L.441-447), especially CBM (condition-based maintenance) for factory machines.

Reviewer 4 Report

Comments and Suggestions for Authors

In this paper, authors introduced a data poisoning attack on an anomaly detection system. By tampering the MEMS accelerometer via acoustic waves, authors showed that the detection system cannot detect the abnormal state. Experimental results were provided detailly. In my opinion, this work contributes to the knowledge of physical attacks on sensors. There are some points should be considered.

Please explain the novelty of this paper with the published works such as the previously data poisoning attack against an on-device learning anomaly detector [8].

Please describe a little bit about the experimental setup. “The measurement values of the accelerometer are recorded via a controller (Raspberry Pi Pico in Sec. 6 and Raspberry Pi4 Model B in Sec. 7).” => How is the data of the accelerometer transmitted?

How did authors selected the value of the error threshold?

In this work, authors use autoencoder and ELM. I wonder if authors can use other kinds of networks to overcome this kind of physical attack.

Authors should check the paper carefully and correct minor errors along the paper. For example, in Figs. 11-13, authors should write “frequency“ instead of “freqency”.

Comments on the Quality of English Language

Minor editing of English language required.

Author Response

Thank you very much for providing important comments. We are very grateful to you for spending time on our paper. We list the responses to your comments below.  

Comments 1: Please explain the novelty of this paper with the published works such as the previously data poisoning attack against an on-device learning anomaly detector [8].

Response 1: We apologize for the difference from our previous work not being described clearly. The main contribution of this paper is targeting a more practical anomaly detection system than the one in the previous report. Specifically, the concept drift detection and multi-instance prediction have been applied and we have shown the new threats caused by these new features. We modified the description to clarify the difference from the previous paper (L.59-61, 246-253, 275-277).  

Comments 2: Please describe a little bit about the experimental setup. “The measurement values of the accelerometer are recorded via a controller (Raspberry Pi Pico in Sec. 6 and Raspberry Pi4 Model B in Sec. 7).” => How is the data of the accelerometer transmitted?

Response 2: We apologize that the measurement setup was not described enough. We used different controllers in each experiment, but the settings were the same during data collection. The accelerometer was connected to controllers (Raspberry Pi) via the I2C bus. The collected data was sent to a computer via the serial bus and used for evaluation. Although previous studies have shown that all anomaly detection algorithms in this paper can be run in real-time on these Raspberry Pis, to ensure the reproducibility of this experiment, we performed evaluations on the computer using data acquired in advance. We added descriptions about missing parts of the experimental setups (L.292-295, 297-298, 300-301).  

Comments 3: How did authors selected the value of the error threshold?

Response 3: We manually selected the value for the sake of convenience in this paper. We added an explanation of the threshold value selection (L.379-381).  

Comments 4: In this work, authors use autoencoder and ELM. I wonder if authors can use other kinds of networks to overcome this kind of physical attack.

Response 4: It is possible that similar attacks could be performed in principle even if different types of NNs are employed; this is because this attack is performed via a malicious instance (model) in the anomaly detector. Due to limited computational resources in current edge devices, it is difficult to realize an anomaly detection system such as the one used in this paper without using lightweight NNs such as ELM. However, even if larger and more advanced NNs can be employed in the future, edge AI systems will need countermeasures against such attacks themselves. We added the explanation (L. 421-426, 441-447).  

Comments 5: Authors should check the paper carefully and correct minor errors along the paper. For example, in Figs. 11-13, authors should write “frequency“ instead of “freqency”.

Response 5: Thank you for pointing out the mistake. We have made some spelling mistakes in Figs. 11-13 and other corrections.  

Round 2

Reviewer 1 Report

Comments and Suggestions for Authors

Comments 1: As stated in the paper, the attacker exploits the multi-instance feature of the detector, in the Attack procedure section of the paper it says that for simplicity only one normal state is acceptable, but in the experimental part there is also only one normal state, more experiments with multiple normal states seems to be more reasonable.

Comments 2: The figures in the experimental part of the article are blurry, and it is recommended to use vectorial image format.

Author Response

Thank you very much for providing important comments.  

Comments 1: As stated in the paper, the attacker exploits the multi-instance feature of the detector, in the Attack procedure section of the paper it says that for simplicity only one normal state is acceptable, but in the experimental part there is also only one normal state, more experiments with multiple normal states seems to be more reasonable.

Response 1: As you mentioned, we assumed that there is only one normal state in this experiment. The anomaly detector works even when there are multiple normal states, and the attack shown in this paper also works. It is because the number of normal states (instances) does not affect the attack; the attack only uses an additional malicious instance, which is separated from normal instances. Therefore, we choose an experimental setup with only one normal state to simplify the scenario and focus on the outcome of the attack. To clear this for readers, we added descriptions (L.436 -440).  

Comments 2: The figures in the experimental part of the article are blurry, and it is recommended to use vectorial image format.

Response 2: Thank you for pointing out the problem with the figures. We have replaced Figs. 10-13, 15 and 16 with higher quality ones.